# Impact of the early COVID-19 pandemic on outcomes in a rural Ugandan neonatal unit: A retrospective cohort study

**Anna Hedstrom**[1]*, **Paul Mubiri**[2], **James Nyonyintono**[3], **Josephine Nakakande**[3], **Brooke Magnusson**[4], **Madeline Vaughan**[4], **Peter Waiswa**[2], **Maneesh Batra**[1]

**1** Departments of Pediatrics and Global Health, Neonatology, Seattle Children's Hospital, University of Washington, Seattle, WA, United States of America, **2** Makerere University School of Public Health, Mulago, Kampala, Uganda, **3** Kiwoko Hospital, Luweero Nakaseke, Kiwoko, Uganda, **4** Adara Development, Rozelle, NSW, Australia

* hedstrom@uw.edu

## Abstract

### Background

During the early COVID-19 pandemic travel in Uganda was tightly restricted which affected demand for and access to care for pregnant women and small and sick newborns. In this study we describe changes to neonatal outcomes in one rural central Ugandan newborn unit before and during the early phase of the COVID-19 pandemic.

### Methods

We report outcomes from admissions captured in an electronic dataset of a well-established newborn unit before (September 2019 to March 2020) and during the early COVID-19 period (April–September 2020) as well as two seasonally matched periods one year prior. We report excess mortality as the percent change in mortality over what was expected based on seasonal trends.

### Findings

The study included 2,494 patients, 567 of whom were admitted during the early COVID-19 period. During the pandemic admissions decreased by 14%. Patients born outside the facility were older on admission than previously (median 1 day of age vs. admission on the day of birth). There was an increase in admissions with birth asphyxia (22% vs. 15% of patients). Mortality was higher during COVID-19 than previously [16% vs. 11%, p = 0.017]. Patients born outside the facility had a relative increase of 55% above seasonal expected mortality (21% vs. 14%, p = 0.028). During this period patients had decreased antenatal care, restricted transport and difficulty with expenses and support. The hospital had difficulty with maternity staffing and supplies. There was significant community and staff fear of COVID-19.

**Data Availability Statement:** All relevant data are within the manuscript and its Supporting Information files.

**Funding:** Data collection and data analysis (PM) was supported by Adara Development (https://www.adaragroup.org/). Adara development did not influence the study design, data analysis/interpretation, information reported nor decision to publish.

**Competing interests:** The authors have declared that no competing interests exist.

## Interpretation

Increased newborn mortality during the early COVID-19 pandemic at this facility was likely attributed to disruptions affecting maternal and newborn demand for, access to and quality of perinatal healthcare. Lockdown conditions and restrictions to public transit were significant barriers to maternal and newborn wellbeing, and require further focus by national and regional health officials.

## Background

The COVID-19 pandemic has stressed health systems around the world. In addition to the massive number of deaths due to infection with the SARS-CoV-2 virus, the pandemic and associated control measures have had a negative impact on primary and essential health care services due to disruptions in health workforce and supply chains, overwhelmed health facilities and decreased care seeking for non-COVID-19 causes [1]. This has resulted in substantial negative effects on human health around the world, particularly among the most vulnerable populations. Much of the progress towards reducing maternal and newborn deaths as well as stillbirth over the last several decades has necessitated health systems strengthening, increasing care seeking and optimizing supply chains for commodities relevant to mothers and babies [2, 3]. Progress towards achieving the Sustainable Development Goals and Every Newborn Action Plan targets for reducing maternal deaths, newborn deaths and stillbirths is expected to slow substantially due to COVID-19 impacts, and may even regress without significant adjustments [3–7]. Therefore, understanding and documenting the indirect effects of the COVID-19 pandemic on newborn health and access to care are crucial to design strategies to address vulnerabilities in the healthcare system.

While direct effects of SARS-CoV-2 infection among mothers or newborns has not been reported as a significant contributor to the overall mortality due to COVID-19, evidence for impact on maternal and newborn services such as access to facility-based deliveries, antenatal care and emergency obstetric care is emerging [8–10]. An early survey by the WHO reported 80% of countries had disruptions to essential health services and these were exacerbated in lower income countries [1]. Little has been published, however, on availability of peripartum care or neonatal outcomes from low resource facilities during the COVID-19 pandemic [11–14].

Over the past decades, Uganda has made significant strides in reducing under-five mortality and maternal deaths; however as of 2019, neonatal mortality had plateaued at 20 deaths per 1000 live births [15]. Twenty five percent of births in Uganda are not delivered by skilled birth attendant, post-natal services remain low and geographical inequalities in access to maternal services exist [16]. It is in this background in which the first case of COVID-19 in Uganda was reported on 21 March 2020 [17]. As a result, the government of Uganda instituted measures to control the spread of the disease including restrictions to movement and a national lockdown that began 30 March and began to ease on 26 May. These restrictions exacerbated existing maternal and newborn service challenges, most notable in decreased ability for pregnant women or parents with a newborn to travel urgently to health facilities. As of May 2021, Uganda had recorded over 43,000 cases of COVID-19 with 357 deaths [18]. Pandemic control measures affecting access to newborn care remain highly dynamic and no country-level data of neonatal survival during this period are available.

We have been prospectively collecting data at one rural Ugandan neonatal unit since 2005 where provision of care for small and sick newborns, including CPAP therapy has been well

established. During the early COVID-19 pandemic, this facility was able to obtain personal protective equipment for staff and maintain neonatal care but peripartum mothers had difficulty traveling to the facility due to lockdown. The aim of this study is to describe this facility's experience with providing peripartum care during the pandemic by utilizing this longitudinal dataset to explore changes to facility-based neonatal outcomes before and during the early phase of the COVID-19 pandemic.

## Methods

### Study design

Retrospective cohort study of a clinical database.

### Study population

Included in this study are patients admitted to the Kiwoko Hospital (KH) neonatal unit during the period of study. There were no exclusion criteria.

### Setting

Kiwoko Hospital is a rural, private, not-for-profit secondary level care hospital that acts as a referral center for three districts (total population 1,000,000) in central Uganda. During fiscal year 2019–2020, the perinatal mortality rate in the hospital's district (Nakaseke) was in the range of 29–42 per 1,000 live births and 19–28 in the two other districts the hospital serves (Nakasongola and Luwero) [19, 20]. The neonatal mortality rate for Uganda was 27 per 1,000 live births [21].

The KH neonatal care unit was established in 2001 and is a regional leader in care of small and sick newborns [22]. The unit admits more than 1,000 patients annually, generally neonates with a gestational age greater than 24 weeks and up to 44 weeks. About half of admitted patients are born at KH and the remainder are "outborn," or admitted after birth at home or another facility. Outborn patients are treated in the same unit and receive the same care as those born at KH. There are 5–7 nurses on duty each shift, as well as one assigned medical officer and one pediatric physician who round on the patients and are on call each day. Staff generally live on site. Electricity is continuously available with the help of a standby generator. The unit provides thermoregulatory support primarily from radiant warmers and incubators, infection control and treatment, nasogastric and cup feeding, intravenous hydration, phototherapy, blood transfusion, basic laboratory services, oxygen therapy, and pulse oximetry. Improvised bCPAP has been the standard of care for respiratory failure in the unit since 2012 and is assembled using donated RAM nasal cannulas [23]. The unit does not provide surfactant, mechanical ventilation, total parenteral nutrition nor therapeutic hypothermia for birth asphyxia. The national referral hospital is located two hours travel by car and referrals from the Kiwoko NICU are transported free of charge. Admissions via ambulance to KH are assisted by government primary health care subsidies. General care in the neonatal unit is subsidized and families pay for any additional lab tests or imaging. Mothers are offered free accommodation and basic meals during their baby's stay.

### COVID-19 experience

The first COVID-19 tests at Kiwoko hospital were conducted on 5 May 2020 and the first case of COVID-19 was confirmed on 2 August. Following a total of 752 tests of staff, patients and community members, 14 total cases were confirmed before October and transferred to isolation centers. No neonates were denied care and access to and provision of care continued

unchanged in the neonatal unit. Mothers were screened for symptoms on entrance to the unit. Experience at KH was determined through interviews with neonatal and maternity unit doctors as well as supervising midwives and nurses (summarized in S1 Table).

Nationally during the 2020 period of the COVID-19 pandemic in Uganda, restrictions included: banning mass gatherings (18 March), closing schools (20 March), suspending public transportation and requiring police presence for private transport (25 March), and a nation-wide lockdown with curfew (30 March). All non-essential services and activities were closed in this period [24]. Until 19 April, approval from a Resident District Commissioner was required to move during a medical emergency, and each district had only one Commissioner [25, 26]. Transport restrictions began to ease on 26 May: public transport resumed at half capacity (and increased cost) on 4 June, and the most common form of transport (via motorcycle) resumed on 27 July. Overnight curfews remained in place through the end of 2020.

## Dataset

The neonatal unit has maintained an electronic database of all neonatal admissions since late 2012. On admission, nurses record patient and maternal information on a designated bedside form. After discharge, a data entry team extracts additional information from the medical file including treatments and final diagnosis as assigned by the physician at discharge, death, or transfer. Data are manually entered into Epi Info version 7 [27]. The data was exported to STATA v.15 for analysis [28].

## Outcome

We utilized this database to explore outcomes from all infants admitted to the neonatal unit during the pre-COVID-19 period (October 2019 through March 2020) and the first six months of the pandemic in Uganda, which we will refer to as the early COVID-19 period, or the COVID-19 period (April through September 2020). These periods were selected to capture the first lockdown period in Uganda (April/May) as well as the subsequent phased relaxation of COVID-19 restrictions (June/July) and then two additional months (August/September) to allow for six month seasonal comparisons with previous years. We also show seasonal variation and pre-existing trends in mortality by reporting outcomes from reference periods one year prior to study periods: Oct 2018 –March 2019 and April–Sep 2019. Death before discharge was the primary outcome- defined as a death as an infant that was admitted to the neonatal unit and died before discharge home.

## Other variables

The analysis was further stratified by inborn/outborn status. Patients were classified as 'inborn' if they were born at KH while others were classified as 'outborn' (born at another facility, at home or on the way to the hospital). Birthweight-specific and diagnosis-specific mortalities were also computed. When birthweight was unknown, admission weight was used if the patient was admitted within 3 days of birth. Gestational age is not reliably available from this dataset and is therefore not reported for this study. Primary diagnoses were assigned by doctor at patient discharge, transfer, or death. Additional data on availability of health workforce in the neonatal unit was retrospectively reported by the nurse in-charge.

## Statistical analysis

The mortality rate pre and during the early COVID-19 period was computed as a proportion of infants who died to the total number of infants admitted to the unit in the same period. To

determine the potential impact of the COVID-19 pandemic on neonatal mortality in the unit, we computed the relative change in mortality rate between two time periods of the same months for the 2 sequential years. Proportion tests were used to compare the rates between two time periods of the same months.

### Ethical considerations

Human Subjects Approval was obtained from Makerere University School of Public Health Institutional Review Board (protocol number 917) and approved by the Uganda National Council for Science and Technology (registration number SS813ES). The data were fully anonymized before accessed by the research team and the ethics committee waived the requirement for informed consent. The University of Washington institutional review board designated this as an exempt study.

## Results

The study included 2,494 patients admitted during the period of study and admissions during each period are shown in Table 1. A total of 567 patients were admitted during the early COVID-19 period (April-Sept 2020) representing a 14% decrease in admissions compared to the same period in the preceding year (April-Sept 2019). Out of 2,494 patients included in the analysis, 49% (1,234) were outborn (born elsewhere and admitted to the KH unit). Proportion of outborn patients decreased during the early COVID-19 period, as shown in Fig 1. Five patients did not have inborn/outborn status recorded, and these were excluded from the stratified analyses by birth location. The average daily census increased during the early COVID-19 period to 42.3 (sd 6.3) from a range of 38.6–41.6 in the periods before COVID-19.

Maternal and neonatal demographics are shown in Table 2. Nineteen percent (474/2,494) of patients had missing birth weight and admission weight was used as proxy for 400 of this group.

As seen in Table 2, during the early COVID-19 period admitted patients were born to younger mothers (median of 24 vs. 25 years). Both expectant mothers and newborns were more likely to be transported via ambulance or on foot during COVID-19.

Outborn patients during the early COVID-19 period were admitted on average one day later than pre-COVID-19 (median admission at 1 day of age vs. on the day of birth). Outborns admitted were less likely to have been born at home (9.5% vs 11.5%), and their births were less likely to be attended by a doctor (9% vs. 11%) (Table 2).

There was an increased proportion of admission with birth asphyxia, most notably among outborn patients (28% from 17% pre-COVID-19). Concurrently, outborn admissions were less likely to have a diagnosis of prematurity (40% down from 45%) or infection (21% from 26%). Inborn admissions had an increased proportion due to low birthweight (58% from 54%) and very low birthweight (17% from 11%) while these decreased among outborn patients (Table 2).

**Table 1. Admissions per period, unit census and staffing levels.**

|  | Reference periods | | pre-COVID-19 period | Early COVID-19 period |
|---|---|---|---|---|
| Date range | Oct 2018 –Mar 2019 | April–Sep 2019 | Oct 2019 –Mar 2020 | April–Sep 2020 |
| Number of patients admitted (n) | 649 | 659 | 619 | 567 |
| Outborn n (%) | 326 (50.1%) | 323 (49.1%) | 321 (51.6%) | 264 (46.1%) |
| Average Daily Census (sd) | 38.6 (4.9) | 41.3 (6.6) | 41.6 (5.3) | 42.3 (6.3) |
| Nurses per shift | 5 | 5 | 5 | 5 |

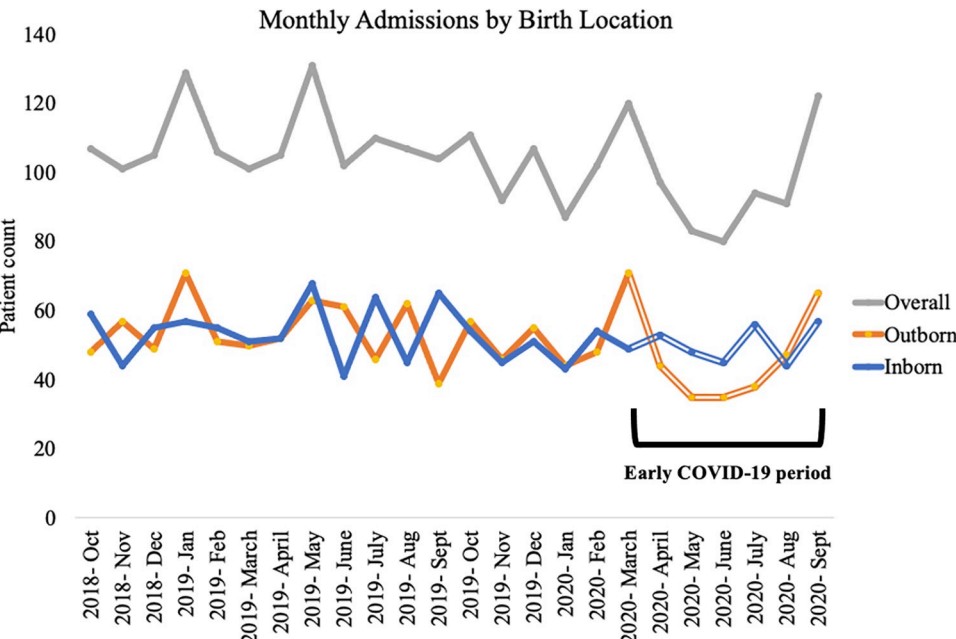

**Fig 1. Admissions by birth location.** Proportion of admissions of outborn patients decreased to 46.1% during the early COVID period (April–September 2020) from 51.6% pre-COVID (October 2019 to March 2020). All transport was banned or tightly limited during April and May and public transport was costly and restricted during June and July. "Outborn" refers to patients born outside KH. "Inborn" includes those born at KH.

All patients were more likely to be treated with phototherapy (44% up from 34%) and less likely to receive a blood transfusion (3% down from 5%). Bubble CPAP therapy, however, was used more frequently in only inborn patients (17% vs. 14%) during the pandemic and likely corresponds to the increase in prematurity among this group (Table 2).

Mortality significantly differed between the COVID-19 period and pre-COVID-19 periods [15.7% (89/567) vs. 11.1% (69/619), p = 0.017]. Fig 2 illustrates mortality pre-COVID-19 and during the early COVID-19 period compared with the reference periods one year prior to allow for seasonally appropriate trends. When we compared mortality pre-COVID and the reference period of the same months, mortality increased proportionally by 5.7% while during COVID, mortality increased proportionally by 41.4% compared to the reference period. Therefore an excess mortality of 35.7% (41.4–5.7%) occurred during the early COVID-19 period.

Among the inborn patients, mortality pre-COVID was 7.8 per 100 hospital admissions while during COVID, mortality was 10.9 per 100 hospital admissions. Mortality increased by 20.0% pre-COVID when compared to the reference period, however during COVID, mortality increased by 25.3%. Therefore an excess mortality of 5.3% (25.3% - 20.0%) among inborn patients occurred during the early COVID-19 period.

Among outborn patients, the excess mortality was 55.2% (55.9- -0.7%). Further, mortality among the outborn infants significantly differs during early COVID compared to the immediate pre-COVID period (21.2% vs. 14.3%, p = 0.028).

Table 3 shows increased mortality during the COVID-19 period among almost all categories of diagnosis and birthweight. Mortality among inborn patients with prematurity increased from 8.7% pre to 13.7%. Mortality among the smallest inborn patients (<1.5kg birthweight) was increased from 26.5% to 38.5% while other categories of birthweight among inborns remained unchanged. Outborn patients, however, had significantly increased mortality in all major categories of birthweight- patients <1.5k up through 4kg.

**Table 2. Maternal and neonatal demographics and clinical characteristics during pre-COVID (October 19 –March 20) and early COVID periods (April–September 2020) by birth location.**

| | All Patients | | Inborn Patients | | Outborn Patients | |
|---|---|---|---|---|---|---|
| | pre-COVID (n = 619) % | COVID (n = 567) % | pre-COVID (n = 296) % | COVID (n = 303) % | pre-COVID (n = 321) % | COVID (n = 264) % |
| **Maternal Demographics** | | | | | | |
| **Median maternal age (years, IQR)** | 25 (20–29) | 24 (20–29) | 25 (21.5–29) | 24 (21–30) | 24 (20.29) | 23 (20–28) |
| Antenatal care visits (mean, SD) | 2.6 (1.3) | 2.6 (1.3) | 2.6 (1.3) | 2.6 (1.3) | 2.5 (1.2) | 2.5 (1.3) |
| **Mode of transport to facility:** | | | (pregnant mother) | | (newborn) | |
| Motorcycle | 51.8 | 44.8 | 55.7 | 50.5 | 48.0 | 38.3 |
| Taxi/Special hire | 29.6 | 27.9 | 30.7 | 28.4 | 29.3 | 27.3 |
| Ambulance | 15.8 | 24.7 | 9.5 | 17.5 | 21.8 | 32.9 |
| Bicycle or foot | 0.8 | 1.8 | 1.0 | 2.3 | 0.6 | 1.1 |
| Unknown | 1.9 | 0.9 | 3.7 | 1.3 | 0.3 | 0.4 |
| **Lives outside district** | 67.2 | 67.2 | 64.5 | 65.3 | 69.8 | 69.3 |
| **Type of birth attendant** | | | | | | |
| Doctor | 27.1 | 27.0 | 44.3 | 42.6 | 11.2 | 9.1 |
| Midwife/Nurse | 65.3 | 66.7 | 55.7 | 57.4 | 74.4 | 77.3 |
| Traditional birth attendant | 1.8 | 1.9 | - | - | 3.4 | 4.2 |
| Family member | 4.0 | 2.8 | - | - | 7.5 | 6.1 |
| Unknown/other | 1.8 | 1.6 | - | - | 3.4 | 3.4 |
| **Birth location** | | | | | (n = 585) | |
| Another facility | - | - | - | - | 86.3 | 88.6 |
| Home | - | - | - | - | 11.5 | 9.5 |
| On the way to hospital | - | - | - | - | 2.2 | 1.9 |
| **Cesarean section** | 25.8 | 27.0 | 42.9 | 41.9 | 10.0 | 9.8 |
| **Singleton birth** | 82.2 | 82.4 | 81.4 | 79.9 | 82.9 | 85.2 |
| **Infant characteristics** | | | | | | |
| **Female** | 45.4 | 43.9 | 46.6 | 43.6 | 44.6 | 44.3 |
| **Age at admission (days)** | - | - | - | - | 0 (0–2) | 1 (0–2) |
| Median (IQR) | | | | | | |
| Mean (SD) | - | - | - | - | 2.5 (5.6) | 3.2 (8.8) |
| Low birthweight (<2.5kg) | 51.5 | 51.8 | 53.7 | 58.1 | 49.5 | 44.7 |
| Very low birthweight (<1.5kg) | 14.4 | 15.7 | 11.5 | 17.2 | 17.1 | 14.0 |
| **Primary diagnosis** | | | | | | |
| Prematurity/LBW | 47.7 | 46.9 | 50.3 | 52.8 | 45.2 | 40.1 |
| Birth asphyxia | 15.0 | 22.4 | 12.2 | 17.5 | 17.4 | 28.0 |
| Infection | 21.8 | 18.9 | 17.6 | 17.2 | 25.9 | 20.8 |
| Other | 15.5 | 11.8 | 19.9 | 12.6 | 11.5 | 11.0 |
| **Clinical Course** | | | | | | |
| **Therapies received:** | | | | | | |
| Phototherapy | 34.3 | 44.3 | 30.1 | 42.6 | 38.1 | 46.2 |
| Blood transfusion | 4.8 | 2.5 | 3.7 | 1.6 | 5.9 | 3.4 |
| Bubble CPAP | 14.0 | 15.7 | 13.8 | 16.5 | 14.3 | 14.8 |

▯Yellow highlighted cells show trends of particular clinical significance.

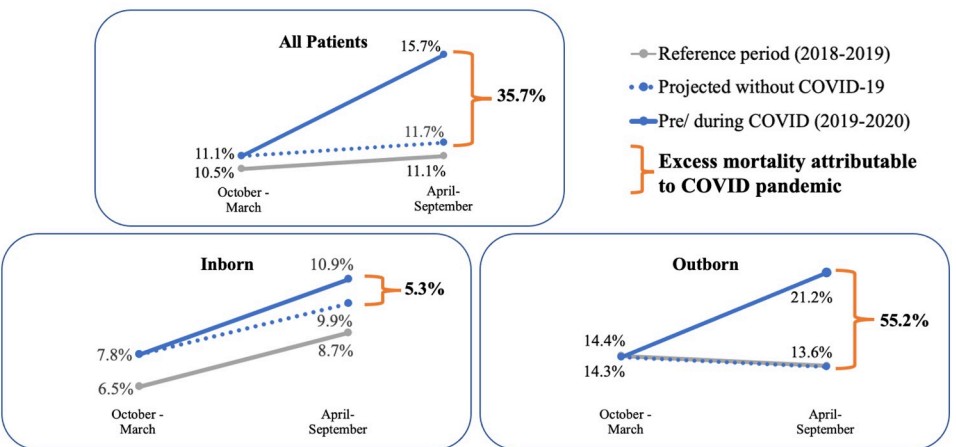

**Fig 2. Newborn mortality by birth location including expected mortality trend (dotted lines) without the COVID-19 pandemic based on seasonally matched periods prior to COVID-19.** Excess mortality is the amount of mortality seen above what would be expected for trend, calculated as the relative difference between periods divided by mortality in the reference period. 1,234 patients were outborn (born outside Kiwoko hospital) and 1,260 born at Kiwoko (inborn) during the periods of study.

Table 4 summarizes the impact of reported maternal child health impacts globally and how they affected KH. Descriptions and data from KH on these topics are found in S1 Table.

## Discussion

Mortality and admissions with birth asphyxia increased while outborn admissions decreased in this rural Ugandan neonatal unit during the first six months of the COVID-19 pandemic. The pandemic was associated with a relative increase in mortality by 36% and 55% excess mortality above seasonal trends among outborn patients. This increased mortality is likely attributed to disruptions due to the pandemic affecting maternal and newborn demand for, access to and quality of peripartum healthcare [45].

**Table 3. Mortality by birth location, birthweight and diagnosis during early COVID and pre-COVID periods.**

| | Inborn Patients | | Outborn Patients | |
|---|---|---|---|---|
| | pre-COVID period (Oct 19—Mar 20) n = 296 | Early COVID period (April–Sep 20) n = 303 | pre-COVID period (Oct 19—Mar 20) n = 321 | Early COVID period (April–Sep 20) n = 264 |
| **Birth Location** | | | | |
| | (23/296) 7.8% | (33/303) 10.9% | (46/321) 14.3% | (56/264)* 21.2% |
| **Diagnosis** | | | | |
| Prematurity/ low birthweight | (13/149) 8.7% | (22/160) 13.7% | (18/145) 12.4% | (15/106) 14.1% |
| Birth asphyxia | (5/36) 13.9% | (8/53) 15.1% | (19/58) 33.9% | (25/69) 35.1% |
| Infection | (0/52) 0% | (1/52) 1.9% | (8/83) 9.6% | (8/55) 14.5% |
| **Birthweight** | | | | |
| <1.5 kg | (9/34) 26.5% | (20/52) 38.5% | (12/55) 21.8% | (13/37) 35.1% |
| 1.5 to 2.49 kg | (6/125) 4.8% | (5/124) 4.0% | (6/104) 5.8% | (10/81) 12.3% |
| 2.5 to 4kg | (8/126) 6.3% | (8/119) 6.7% | (21/131) 16.0% | (28/122) 23.0% |
| >4 kg | (0/11) - | (0/8) - | (1/5) 20.0% | (2/12) 16.7% |

*p< 0.05. ▯Yellow highlighted cells show trends of particularly clinical significance.

**Table 4. Summary of impacts on maternal child and health reported globally during the early COVID-19 pandemic and the experience at Kiwoko hospital.**

| Kiwoko Impacted | Kiwoko not impacted |
|---|---|
| Maternity | |
| Restricted transport for mothers[1, 29, 30] | Increased adolescent pregnancy[31, 39] |
| Decreased antenatal care[1, 31–34] | Increased stillbirth[2, 32, 39] |
| Decreased facility birth[1, 6, 8, 12, 30, 33–35] | Decreased cesarean sections[39] |
| Increased births at home or with traditional birth attendant[36] | Restricted personal protective equipment (PPE)[1, 8, 30] |
| Decreased availability of labor medications [32, 37] | |
| Decreased staffing levels[1, 3, 8, 38] | |
| Neonatal Unit | |
| Restricted transport for babies[1, 29, 30] | Decreased staffing levels[1, 3, 8, 38] |
| Decreased neonatal intensive care admissions[14, 32, 40] | Restricted supplies/ personal protective equipment (PPE)[1, 8, 30] |
| Decreased outborn admissions[8, 41] | Decreased kangaroo mother care (KMC)[8, 44] |
| Increased preterm birth[14, 32] | |
| Increased birth asphyxia[14] | |
| Decreased blood supply[1] | |
| Increased facility neonatal mortality[12, 40, 42] | |
| Decreased facility-based infant follow-up[8, 43] | |
| Parents | |
| Decreased maternal support by family at hospital[8] | |
| Difficulty with medical expenses/ food security[1, 24, 31, 40] | |
| Staff and Community | |
| Staff COVID-19 fear[3, 8] | |
| Community fear of COVID-19 at facilities[8, 33, 34] | |

We also found that the early COVID-19 period was associated with fewer admissions but increased daily census suggesting longer lengths of stay. Inborn patients were more likely to be preterm or very low birthweight and be treated with bubble CPAP. Outborn admissions decreased, were older on admission and were less likely to be delivered by a doctor. All patients were more likely to have birth asphyxia and be treated with phototherapy. These finding suggest increased patient acuity and aligns with resultant increased mortality during this period.

In Uganda, lockdown in April and May and restricted, expensive public motorcycle transit through July created significant barriers to transport that exacerbated existing delays to maternal and neonatal care, including delayed decision to seek care and prolonged time to reach to care [41, 45]. Increased mortality among newborns in this setting due to the indirect effects of the pandemic could quickly dwarf the direct mortality rate from COVID-19 infections, and likely did so in 2020. Although primary data is sparse, other sites have also reported increased facility-based neonatal deaths during COVID-19 lockdown periods, especially among outborn patients [12, 14, 42]. Heterogeneity in mortality among sites suggests the indirect impacts of COVID-19 are context-specific [41]. The experience at KH is described in Table 4 and S1 Table. KH was resourced to provide sufficient personal protective equipment to staff and parents in the neonatal unit, was able to adequately space mothers staying in on-campus quarters, and isolated patients demonstrating COVID-19 symptoms. This ability to respond to the pandemic beyond baseline newborn resources and outcomes at KH that exceed many other facilities in Uganda, may explain why outcomes could have been worse elsewhere.

Proportion of admissions due to prematurity was increased among inborn babies but not among outborns. With single facility data we cannot infer population-level changes in prematurity rates during the pandemic as other regions have, but decreased admissions of outborn premature infants suggests they may have died before reaching KH or were less likely to be transferred from other facilities [2, 11]. Mothers who did deliver at KH may have been triaged from the community for their risk of preterm delivery while others delivered elsewhere. Despite their birth location, the smallest patients (<1.5kg) had increased mortality which reveals the fragile survival of these patients at baseline.

Limitations to this study include the fluidity and dynamic nature of impacts of the early pandemic on demand for, access to and provision of quality maternal and newborn care services. For example, transportation impacts were greatest early in the COVID-19 study period due to lockdown restrictions, but further impacts on quality of care in the hospital may have been later in the period when COVID-19 was diagnosed locally and affected maternity staffing, community morale and supply chain of obstetric medications. This analysis does not capture subsequent months during the COVID-19 pandemic, including Uganda's peak in infections in December 2020 and most recent wave in June 2021 [46]. Finally, we could not account for all impacts on maternal and newborn health, as impacts to antenatal care may impact the health of the baby independent of the neonatal care, either during the newborn period or later in life.

Strengths of the study include the stable care model of the KH neonatal unit and the established dataset with ability to make comparisons with reference years to account for seasonal trends.

COVID-19-related disruptions to the provision of child health service and outcomes, including the impact on immunizations and primary health care, have been well documented [39, 43]. This study is among few reporting impact to essential maternal/newborn care including access to safe delivery and transfer to neonatal inpatient care [8, 11–14, 32, 40, 47]. We report the pandemic's disruptions to outcomes for small and sick newborns, who are among the most vulnerable humans and accounted for one fifth of childhood deaths pre-pandemic [15]. The World Health Organization has highlighted the importance of evaluation of health care vulnerabilities and modifications that will allow the continuation of essential health services during the pandemic [48]. Reducing impacts on vulnerable newborns in the continued pandemic will take coordinated and deliberate actions by policymakers, countries, and regional public health officials to ensure that access and quality to maternal and newborn health services are prioritized and optimized.

## Supporting information

**S1 Table. Impacts on maternal child and health reported globally during the early COVID-19 pandemic and their experience at Kiwoko hospital.**
(DOCX)

**S2 Table. Variable definitions.**
(XLSX)

**S1 Dataset.**
(XLSX)

## Acknowledgments

Sisters Immaculate Nakku and Hajara Nabunya, Heidi Nakamura, Drs. Becca Jones and Mushin Nsubuga.

## Author Contributions

**Conceptualization:** Anna Hedstrom, Paul Mubiri, James Nyonyintono, Madeline Vaughan, Maneesh Batra.

**Formal analysis:** Anna Hedstrom, Paul Mubiri.

**Investigation:** Anna Hedstrom.

**Methodology:** Anna Hedstrom, Brooke Magnusson, Madeline Vaughan, Peter Waiswa, Maneesh Batra.

**Project administration:** Anna Hedstrom, James Nyonyintono, Josephine Nakakande, Brooke Magnusson.

**Supervision:** Anna Hedstrom, Josephine Nakakande, Madeline Vaughan, Peter Waiswa, Maneesh Batra.

**Validation:** Anna Hedstrom, Paul Mubiri.

**Visualization:** Anna Hedstrom, Paul Mubiri.

**Writing – original draft:** Anna Hedstrom, Paul Mubiri.

**Writing – review & editing:** Anna Hedstrom, Paul Mubiri, James Nyonyintono, Josephine Nakakande, Brooke Magnusson, Madeline Vaughan, Peter Waiswa, Maneesh Batra.

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
