## [Decision Letter · Decision Letter 0]

14 Jul 2021

PONE-D-21-20950

Outcomes in a Rural Ugandan Neonatal Unit Before and During the Early COVID-19 Pandemic: a Retrospective Cohort Study

PLOS ONE

Dear Dr. Hedstrom,

Thank you for submitting your manuscript to PLOS ONE. After careful consideration, we feel that it has merit but does not fully meet PLOS ONE’s publication criteria as it currently stands. Therefore, we invite you to submit a revised version of the manuscript that addresses the points raised during the review process.

Please address the issues and revise accordingly.

We look forward to receiving your revised manuscript.

Kind regards,

Academic Editor

PLOS ONE

Journal Requirements:

2. In ethics statement in the manuscript and in the online submission form, please provide additional information about the database used in your retrospective study. Specifically, please ensure that you have discussed whether all data were fully anonymized before you accessed them and/or whether the IRB or ethics committee waived the requirement for informed consent. If patients provided informed written consent to have their data used in research, please include this information.

"Data collection and data analysis (PM) was supported by Adara Development (https://www.adaragroup.org/). Adara development did not influence the study design, data analysis/interpretation, information reported nor decision to publish. "

We note that one or more of the authors have an affiliation to the commercial funders of this research study : Adara Development.

3.1. Please provide an amended Funding Statement declaring this commercial affiliation, as well as a statement regarding the Role of Funders in your study. If the funding organization did not play a role in the study design, data collection and analysis, decision to publish, or preparation of the manuscript and only provided financial support in the form of authors' salaries and/or research materials, please review your statements relating to the author contributions, and ensure you have specifically and accurately indicated the role(s) that these authors had in your study. You can update author roles in the Author Contributions section of the online submission form.

3.2. Please also provide an updated Competing Interests Statement declaring this commercial affiliation along with any other relevant declarations relating to employment, consultancy, patents, products in development, or marketed products, etc.  

Please respond by return email with an updated Funding Statement and Competing Interests Statement and we will change the online submission form on your behalf.

Reviewers' comments:

Reviewer's Responses to Questions

**Comments to the Author**

1. Is the manuscript technically sound, and do the data support the conclusions?

Reviewer #1: Yes

Reviewer #2: Yes

2. Has the statistical analysis been performed appropriately and rigorously? 

Reviewer #1: No

Reviewer #2: No

3. Have the authors made all data underlying the findings in their manuscript fully available?

Reviewer #1: Yes

Reviewer #2: No

4. Is the manuscript presented in an intelligible fashion and written in standard English?

Reviewer #1: Yes

Reviewer #2: No

5. Review Comments to the Author

Reviewer #1: Title: Outcomes in a Rural Ugandan Neonatal Unit Before and During the Early COVID-19 Pandemic: a Retrospective Cohort Study (manuscript number: PONE-D-21-20950)

Review by Ermias Sisay Chanie /MSc in pediatrics and Child health Nursing)

Debre Tabor University, Debre Tabor, Ethiopia

Thanks for give an opportunity to plos one chief editors to reviewing this article. This research is important on reducing neonatal mortality and morbidity in worldwide in general and in study ara in particular, in the meantime it is one of the international concerns. African countries including Ugandan works to reduce the mortality of neonate through monitoring and evaluation the ‘’outcomes neonate in Neonatal Unit”.

Minor

1. General English: The review recommends the article be copyedited to improve language and grammar used

2. Check the table number frequency, percentage and p- value for each column

Major concern

Title

1. In my point of view the title better to modified based on your conclusion, since it focuses on the impact of COVID-19 in “maternal and newborn demand for, access to and quality of peripartum healthcare’’ etc.

2. Rural vs Neonatal Unit? I am not clear with it. I am not expected neonatal unit in rural area.

Introduction

1. In my view, better to focus on your objective, which mean the effect of COVID-19 on maternal health, or neonatal health or healthcare system in general

2. Try to show any attempting solution to reduce the burden of COVID-19 related to your objective in the study rea

3. The identified gaps before conducting the study? And the implication of the study to fill the appreciated gaps

Methods

1. Discuss about your study populations vs source population of the study participant clearly

2. your Inclusion criteria vs Exclusion criteria of the study participant clearly

3. Better to included data set in statical analysis and try to explore about descriptive statics that was used etc.…?

4. Adding about goodness of fit and model fitness?

Reviewer #2: Abstract is too long and should not exceed 300 words for The PLOS ONE. Please describe clearly the main objective of this study and summarize the most important results and their significance.

Introduction:

The overall aim of this study and whether that aim was achieved are missing.

Methods:

The role of the funding source should not be in the method section

The statistical analysis to compare the groups are missing or incorrect (proportion tests?)

Results:

Fig 1. It should clarify the term “outborn” and “inborn” in the legend

Table 2. the “+-“ should replace the points in all numbers. Same for the text across the manuscript

p-values should be added to the table. The statistical analysis should be described in the table.

Figure 2 legend should describe the sample size and the statistical analysis used. It should not repeat the results of the figure. It also should specify the term inborn and outborn.

6. PLOS authors have the option to publish the peer review history of their article (what does this mean?). If published, this will include your full peer review and any attached files.

Reviewer #1: No

Reviewer #2: No

---

## [Author Response · Author response to Decision Letter 0]

6 Oct 2021

Response to Reviewers

Reviewer #1: 

Minor

1. General English: The review recommends the article be copyedited to improve language and grammar used

We appreciate this recommendation. Our goal is to be grammatically accurate. The English language/grammar use has been edited where appropriate. If the editors have further suggestions for optimizing grammar we are open to them.

2. Check the table number frequency, percentage and p- value for each column

These have been checked, please see discussion of p values in response to reviewer 2 below.

Major concern

Title

1. In my point of view the title better to modified based on your conclusion, since it focuses on the impact of COVID-19 in “maternal and newborn demand for, access to and quality of peripartum healthcare’’ etc.

The title has been modified to “Impact of the early COVID-19 Pandemic on Outcomes in a Rural Ugandan Neonatal Unit: a Retrospective Study”. Although we agree with the reviewer that the impact on “maternal and newborn demand for, access to and quality of peripartum healthcare” is important we did not have any direct measures of this so do not feel comfortable including it in the title.

2. Rural vs Neonatal Unit? I am not clear with it. I am not expected neonatal unit in rural area.

As the reviewer notes, it is unusual, however Kiwoko hospital is a secondary neonatal unit in a rural area. Due to its clinical reputation, it has become a referral unit for three districts in central Uganda.

Introduction

1. In my view, better to focus on your objective, which mean the effect of COVID-19 on maternal health, or neonatal health or healthcare system in general

The introduction text has been updated to focus on the effect on maternal/neonatal health and the healthcare system in general:

“An early survey by the WHO reported 80% of countries had disruptions to essential health services and these were exacerbated in lower income countries. Little has been published, however, on low resource facility based neonatal outcomes from low resource facilities during the COVID-19 pandemic, nor describing experiences of peripartum care availability.“

2. Try to show any attempting solution to reduce the burden of COVID-19 related to your objective in the study rea

We added to the introduction: “During the early COVID-19 pandemic, this facility was able to obtain personal protective equipment for staff and maintain neonatal care but peripartum mothers had difficulty traveling to the facility due to lockdown.”

3. The identified gaps before conducting the study? And the implication of the study to fill the appreciated gaps

We added “Pandemic control measures affecting access to newborn care remain highly dynamic and no country-level data of neonatal survival during this period are available.”

Methods

1. Discuss about your study populations vs source population of the study participant clearly

We added “Study Population- Included in this study are patients admitted to the Kiwoko Hospital (KH) neonatal unit during the period of study. There were no exclusion criteria.” The “setting” section goes on to describe the type patients admitted to this unit and the typical provision of care in the unit.

2. your Inclusion criteria vs Exclusion criteria of the study participant clearly

See added text in response to #1 above re: inclusion/exclusion criteria.

3. Better to included data set in statical analysis and try to explore about descriptive statics that was used etc.…?

The dataset is now uploaded in accordance with journal requirements. We agree with the reviewer and focused on descriptive statistics in this report. If further suggestions can be considered, please clarify these.

4. Adding about goodness of fit and model fitness?

 We did not use a model in this analysis.

Reviewer #2: 

Abstract is too long and should not exceed 300 words for The PLOS ONE. Please describe clearly the main objective of this study and summarize the most important results and their significance.

Thank you for this guidance. The abstract is now < 300 words. It has been revised to better describe the objective and most important results and their significance.

Introduction:

The overall aim of this study and whether that aim was achieved are missing.

The introduction has been revised to better describe the aim:

“The aim of this study is to describe this facility’s experience with providing peripartum care during the pandemic by utilizing this longitudinal dataset to explore changes to facility-based neonatal outcomes before and during the early phase of the COVID-19 pandemic.” 

Methods:

The role of the funding source should not be in the method section

It has been removed.

The statistical analysis to compare the groups are missing or incorrect (proportion tests?)

As suggested by the reviewer, we used a proportion test to compare the groups. This is described in the methods:

“Statistical analysis 

The mortality rate pre and during the early COVID-19 period was computed as a proportion of infants who died to the total number of infants admitted to the unit in the same period. To determine the potential impact of the COVID-19 pandemic on neonatal mortality in the unit, we computed the relative change in mortality rate between two time periods of the same months for the 2 sequential years. Proportion tests were used to compare the rates between two time periods of the same months. “

Results:

Fig 1. It should clarify the term “outborn” and “inborn” in the legend

The figure legend has been updated with: 

“Outborn” refers to patients born outside KH. “Inborn” includes those born at KH.”

Table 2. the “+-“ should replace the points in all numbers. Same for the text across the manuscript

p-values should be added to the table. The statistical analysis should be described in the table.

- The decimal points have been revised throughout the tables and manuscript

- We appreciate the comment and have discussed among our authors but respectfully disagree with the style of including p values in table 2 which describes demographic characteristics in the two time periods. P values may be misleading in comparing groups from two different periods given the seasonal nature of neonatal mortality and would be at risk of type 2 errors. 

If the editorial team feels a formal comparison of the demographics is important to our manuscript, we would ask to change this table to a comparison of seasonally matched periods (ie comparing April to September 2020 with April to September 2019). However, we feel the most important data to show is immediately prior to the COVID-19 pandemic. Formal appropriate comparisons of mortality adjusting for seasonal trends are later in the manuscript.

Figure 2 legend should describe the sample size and the statistical analysis used. It should not repeat the results of the figure. It also should specify the term inborn and outborn.

- FIgure 2 legend now:

- no longer repeats the results of the figure

- includes “1,234 patients were outborn (born outside Kiwoko hospital) and 1,260 born at Kiwoko (inborn) during the periods of study.“

- includes “Excess mortality is the amount of mortality seen above what would be expected for trend, calculated as the relative difference between periods divided by mortality in the reference period.”

 This has been done.

---

## [Decision Letter · Decision Letter 1]

2 Nov 2021

Impact of the Early COVID-19 Pandemic on Outcomes in a Rural Ugandan Neonatal Unit: a Retrospective Cohort Study

PONE-D-21-20950R1

Dear Dr. Hedstrom,

We’re pleased to inform you that your manuscript has been judged scientifically suitable for publication and will be formally accepted for publication once it meets all outstanding technical requirements.

Kind regards,

Academic Editor

PLOS ONE

Additional Editor Comments (optional):

Reviewers' comments:

Reviewer's Responses to Questions

**Comments to the Author**

1. If the authors have adequately addressed your comments raised in a previous round of review and you feel that this manuscript is now acceptable for publication, you may indicate that here to bypass the “Comments to the Author” section, enter your conflict of interest statement in the “Confidential to Editor” section, and submit your "Accept" recommendation.

Reviewer #1: All comments have been addressed

Reviewer #2: All comments have been addressed

2. Is the manuscript technically sound, and do the data support the conclusions?

Reviewer #1: Yes

Reviewer #2: Yes

3. Has the statistical analysis been performed appropriately and rigorously? 

Reviewer #1: Yes

Reviewer #2: Yes

4. Have the authors made all data underlying the findings in their manuscript fully available?

Reviewer #1: Yes

Reviewer #2: Yes

5. Is the manuscript presented in an intelligible fashion and written in standard English?

Reviewer #1: Yes

Reviewer #2: (No Response)

6. Review Comments to the Author

Reviewer #1: The authors have adequately addressed your comments raised in a previous round of review and you feel that this manuscript is now acceptable for publication.

Thank you very much indeed

Reviewer #2: (No Response)

7. PLOS authors have the option to publish the peer review history of their article (what does this mean?). If published, this will include your full peer review and any attached files.

Reviewer #1: No

Reviewer #2: No

---

## [Editor Report · Acceptance letter]

4 Nov 2021

PONE-D-21-20950R1 

Impact of the early COVID-19 Pandemic on Outcomes in a Rural Ugandan Neonatal Unit: a Retrospective Cohort Study 

Dear Dr. Hedstrom:

I'm pleased to inform you that your manuscript has been deemed suitable for publication in PLOS ONE. Congratulations! Your manuscript is now with our production department. 

Kind regards, 

on behalf of

Dr. Robert Jeenchen Chen 

Academic Editor

PLOS ONE